# Multifactorial Background for a Low Biological Response to Antiplatelet Agents Used in Stroke Prevention

**DOI:** 10.3390/medicina57010059

**Published:** 2021-01-10

**Authors:** Adam Wiśniewski

**Affiliations:** Department of Neurology, Collegium Medicum in Bydgoszcz, Nicolaus Copernicus University in Toruń, Skłodowskiej 9 Street, 85-094 Bydgoszcz, Poland; adam.lek@wp.pl; Tel.: +48-79-0813513

**Keywords:** platelets, molecular pathology, ischemic stroke, clopidogrel, aspirin, resistance, low biological response, platelet reactivity, antiplatelet therapy, platelet function

## Abstract

Effective platelet inhibition is the main goal of the antiplatelet therapy recommended as a standard treatment in the secondary prevention of non-embolic ischemic stroke. Acetylsalicylic acid (aspirin) and clopidogrel are commonly used for this purpose worldwide. A low biological response to antiplatelet agents is a phenomenon that significantly reduces the therapeutic and protective properties of the therapy. The mechanisms leading to high on-treatment platelet reactivity are still unclear and remain multifactorial. The aim of the current review is to establish the background of resistance to antiplatelet agents commonly used in the secondary prevention of ischemic stroke and to explain the possible mechanisms. The most important factors influencing the incidence of a low biological response were demonstrated. The similarities and the differences in resistance to both drugs are emphasized, which may facilitate the selection of the appropriate antiplatelet agent in relation to specific clinical conditions and comorbidities. Despite the lack of indications for the routine assessment of platelet reactivity in stroke subjects, this should be performed in selected patients from the high-risk group. Increasing the detectability of low antiaggregant responders, in light of its negative impact on the prognosis and clinical outcomes, can contribute to a more individualized approach and modification of the antiplatelet therapy to maximize the therapeutic effect in the secondary prevention of stroke.

## 1. Introduction

The appropriate inhibition of platelets is the main goal of secondary prevention of non-cardioembolic ischemic stroke. According to the updated guidelines of stroke management, the use of antiplatelet agents is recommended for every ischemic stroke without cardioembolic background. The selection of an agent should be personalized; however, acetylsalicylic acid (aspirin) and clopidogrel are commonly used for this purpose. In some cases, such as a minor stroke or high risk transient ischemic attack, it is possible to use dual antiplatelet therapy for a period of 21 days, to minimize the risk of early recurrent stroke [1]. The efficacy of antiplatelet therapy is essential to reduce cardiovascular events. 

However, there are some issues that could reduce a patient’s responsiveness to antiplatelet therapy. The most common is a phenomenon referred to as a resistance to antiplatelet agents, which results in a low biological platelet response, leading to an inappropriate platelet inhibition [2]. Low biological responders may present clinical manifestation in the form of recurrent vascular events despite regular drug intake. This is due to the failure of an antiplatelet agent to prevent cardiovascular incidents. There is still a lack of a standardized definition of a low biological response to antiplatelet agents [3]. The prevalence is estimated at 5–65%, which confirms the scale of this phenomenon and its potential significance [4]. However, the mechanisms leading to high on -treatment platelet reactivity are still unclear and remain multifactorial.

The aim of the current review is to establish the data regarding the background of resistance to antiplatelet agents used in the secondary prevention of ischemic stroke and to explain the possible mechanisms.

## 2. Antiplatelet Agents

Aspirin is the most commonly used antiplatelet agent worldwide. The antiplatelet properties are a result of the acetylation of cyclooxygenase-1 at serine 529 leading to the irreversible inhibition of the transformation of arachidonic acid to prostaglandin G2 and, finally, to thromboxane A2 [5]. Thromboxane A2 is one of the most important factors responsible for platelet activation. In contrast to aspirin, clopidogrel is a pro-drug and needs to be biotransformed by the cytochrome-P450-complex-dependent pathway to be active and to achieve antiaggregative properties. Clopidogrel inhibits platelet activation as an adenosine diphosphate (ADP) inhibitor by binding P2Y12 receptors on platelets, which results in reduced ADP-mediated platelet aggregation with the glycoprotein IIb/IIIa (GPIIb/IIIa) complex pathway [6]. 

Other drugs, such as cilostazol, dipirydamol, ticlopidine or triflusal were also accepted for use in the secondary stroke prevention. However, due to the lack of some evidence of their efficacy confirmed in clinical trials, the lack of numerous studies on low biological response and their marginal application in everyday practice, they were excluded from further considerations.

## 3. High On-Treatment Platelet Reactivity

The biological response may be estimated by platelet function testing. The measurement of platelet reactivity, especially aggregation and activation, is the most commonly used for this purpose. Normally, platelet reactivity values remain low, reflecting the effective platelet inhibition by antiplatelet agents. In case of decreased platelet sensitivity, platelet reactivity reaches higher levels, despite the antiplatelet drug intake, indicating the reduced therapeutic effect. The high on-treatment platelet reactivity corresponds to a low platelet response to an agent or a drug failure. As a result, stroke subjects are not protected properly. The phenomenon of high on-treatment platelet reactivity contributes to the greatest extent to the reduced efficacy of the applied antiplatelet treatment. However, the definition of high on-treatment platelet reactivity is not widely accepted and clinical importance remains uncertain. Several assays were developed to evaluate the platelet function. The vast majority is based on the measurements of platelet aggregation. They all have similar platelet sensitivity and specificity and can be used interchangeably. They are treated as equivalent, as none of the devices achieved a significant advantage over others. Their strength lies in the simplicity of performance, automatic measurement, quick results and high repeatability. Their limitation is still the high cost and, above all, the lack of standardization of measurements. The cut- off values of the most common platelet function assays are shown in Table 1 [7,8].

## 4. Factors Contributing to a Low Biological Response to Antiplatelet Agents

### 4.1. Genetic Factors

Genetic disturbances, related to the metabolic pathways of aspirin action, are one of the most common causes (account for about 30%) associated with a low response to aspirin [9]. Variations in the cyclooxygenase-1 genotype (single nucleotide polymorphisms) may result in changes of the active center that could effectively block the acetylation from aspirin, leading to a drug failure. Polymorphisms of cyclooxygenase-2, associated with the hyperexpression of isoforms that are non-sensitive to aspirin, are also reasons of a low response to antiplatelet agent. Other single nucleotide polymorphisms that may reduce the therapeutic effect of aspirin are associated with: the thromboxane the A2 receptor; von Willebrand factor gene, glycoprotein; Ia/IIa, IIa/IIIb, or VI genes; and *P2RY1* or *P2RY12* genes [10]. 

Other genetic disturbances that may affect aspirin resistance involve microRNAs. These gene regulators, involved in the mRNA post-transcriptional translation and degradation processes, may regulate platelet activation and aggregation and the biological efficacy of aspirin. miRNAs, such as miR-230, miR-19b, miR-150, and miR-126, were demonstrated to be responsible fora low response to aspirin [11]. 

Due to the complex process of the bioactivation of clopidogrel, there are many more genetic variations and polymorphisms of genes involved in biotransformation- influenced drug resistance, compared to aspirin [12]. The most common gene polymorphism associated with a low response to clopidogrel is related to cytochrome P450 (CYP) *2C19* variants. This gene is involved in the hepatic metabolism of clopidogrel [13]. Authors demonstrated that carriers with one or more *CYP2C19* loss-of-function alleles (2* or 3*) were characterized by lower levels of the clopidogrel active metabolite in serum leading to reduced antiplatelet responsiveness [14]. Researchers suggested that this genetic variety may be present in up to 32% of subjects and the incidence of high on-clopidogrel platelet reactivity among them may be as high as over 70% [15]. 

Polymorphisms of *CYP3A4,* another gene involved in the oxidative biotransformation of clopidogrel in the liver, particularly loss of the 1G* allele, are considered for risk factors of clopidogrel resistance due to a low concentration of active metabolites [16]. Genetic variants of the *ABCB1* gene, associated with clopidogrel absorption from intestinal cells, especially the TT variant and wild-type CC, have been demonstrated to be correlated with nonresponsiveness [6]. 

Polymorphisms of *P2RY12* and *GPIIIa* genes, encoding receptors related to the biological activity of clopidogrel, have been also described as having an impact on drug failure [17,18]. In addition, there are reports highlighting that gene combinations or multiple interactions play a greater role in the incidence of the phenomenon of a low response to clopidogrel compared with single nucleotide mutation or variants at specific loci [13]. Research also reported that paraoxonase 1 (PON-1) polymorphisms may be associated with a higher risk of a low response to clopidogel. PON-1 is involved in the last stages of the transformation of 2-oxo clopidogrel into its active metabolite. The mutant allele (*PON-1 Q192R*) may decrease its active form and reduce the inhibition of platelets [19].

### 4.2. Alternative Pathways of Platelet Activation

It is estimated that chronic inflammation and oxidative stress may significantly affect the aspirin response, in particular by alternative pathways of platelet activation. The cytokines produced in the inflammatory processes may induce the activation of cyclooxygenase-2 leading to the synthesis of thromboxane-A2. Oxidative stress may result in the hyperproduction of thromboxane-A2 regardless of the arachidonic acid pathways. Catecholamine burst during inflammation and oxidative stress may directly activate platelets that are non-sensitive to aspirin [20]. 

The significant role of catecholamines and alpha2- adrenergic receptors in the variability of biological responses to clopidogrel has been also shown. Human platelets exhibit alpha2- adrenergic receptors that are involved in high residual platelet reactivity and thrombus stabilization. Catecholamines, especially epinephrine, may potentiate the effect of other agonists (in particular, ADP) and initiate different platelets responses from secretion to activation. As a result, increased ADP-induced platelet aggregation contributes to high-on clopidogrel platelet reactivity [21]. On the other hand, a selective blocker of the alpha-2 adrenergic receptor (atipamezol) is considered as a potential option for reducing the frequency of clopidogrel low responses and increasing the inhibitory effect on platelets [22].

### 4.3. Metabolic Syndrome

Certain metabolic disorders are also associated with a low platelet biological response. Cagirci et al. [23] showed that the prevalence of high on- aspirin platelet reactivity is significantly higher in subjects with metabolic syndrome than in the control group (46.9% vs. 20%, respectively). Researches demonstrated, that a persistent thromboxane-dependent pathway of platelet activation exists, which contributes to a low response to aspirin, both in diabetes and hyperglycemia. Similarly, hyperglycation of proteins in platelets may make them less sensitive to aspirin. There is also a competitive action of glucose and aspirin in the processes of non-enzymatic glycosylation and acetylation [24]. Hyperglycemia is a cause of the formation of spontaneous microaggregates in platelets that could be associated with the development of aspirin resistance [25]. 

Hyperlipidemia is another factor that could modify the platelet response to aspirin. Researchers estimated that the increase in the cholesterol content in the composition of the lipid membrane of platelets is associated with their higher sensitivity to agonists and an increased metabolism of arachidonic acid. In dyslipidemia, the immunomodulatory dyad CD40/CD40L interactions increase the prothrombotic properties of platelets and reduce the efficacy of aspirin [26]. Obesity also reduces the aspirin response due to the increase in leptin and secondarily induced hypercoagulability and low concentration levels of drug distribution [27]. Moreover, enhanced lipid peroxidation result in platelet activation in processes that bypass the classical pathway related to the acetylation of cyclooxygenase-1 [28].

Diabetes is considered to be the most important risk factor for vascular events that is strictly associated with a low response to clopidogrel. The mechanism appears to be similar to aspirin. Excessive hyperglycation of platelet proteins and receptors reduces their sensitivity to the effect of the antiplatelet drug. Diabetic subjects also displayed higher levels of P-Selectin, that is associated with increased platelet aggregation [29]. It was reported that biotransformation of clopidogrel is also reduced in diabetics, resulting in the decreased amount of active metabolite [30]. The duration and severity of diabetes mellitus may have a greater influence on the development of resistance to clopidogrel than compared with aspirin [31]. Hyperlipidemia and obesity may influence the G- protein signalling receptors, including P2Y12 receptor for clopidogrel, and modify its expression, leading to higher platelet activation [32]. Pankert et al. [33] demonstrated that among obese subjects with metabolic syndrome the greater heterogeneity and impaired response to clopidogrel was observed, compared to control group, that additionally confirms the significant influence of metabolic disorders on the efficacy of clopidogrel.

### 4.4. Smoking

Authors revealed that smoking may be an important predictor of aspirin resistance. Smoking increases platelet reactivity and decreases the inhibition of thromboxane biosynthesis, which contributes to the reduction in the biological activity of aspirin [34]. Another mechanism may be an elevated values of platelet factor 4, released from platelet alpha-granules during activation, responsible for higher platelet aggregation in current smokers [35].

In contrast to aspirin, current cigarette smoking may be a factor that protects against high-on clopidogrel platelet reactivity (the smoker’s paradox) [36]. Some authors found that smoking was a strong inductor of cytochrome P450 2B6 and 1A2 isoenzymes, which also take an active part in clopidogrel biotransformation [37,38]. As a result of the drug metabolism acceleration, smoking enhances the antiaggregative effect and platelet inhibition of clopidogrel. Nicotine may increase the density and expression of platelet P2Y12 receptors that can make them more sensitive to clopidogrel [39]. Thus, due to the higher percentage of clopidogrel responders and aspirin non-responders, clopidogrel should be considered as the antiplatelet agent of first choice in the prevention of vascular events among smokers.

### 4.5. Chronic Kidney Failure

Chronic kidney failure is also associated with a higher risk of a low biological aspirin response. Studies reported not only a higher frequency of high on-aspirin platelet reactivity among chronic renal failure subjects [40] but also a significant correlation between a low biological response to aspirin and the severity of chronic kidney disease on the basis of the filtration level [41]. This may be due to the modified metabolism of prostaglandins, enhanced concentration of the von Willebrand factor, or increased expression of glycoprotein IIb/IIIa complex in renal impairment.

Similarly to aspirin, the response to clopidogrel is also diminished in patients with chronic kidney disease. Research reported a higher frequency of clopidogrel low responders among subjects with chronic renal failure compared to subjects with normal renal function [36]. The explanation of this finding may be a decreased expression of clopidogrel-metabolizing enzymes in the liver, impaired absorption and transport of the drug, or abnormal release of ADP from platelet granules in chronic kidney disease subjects [42].

### 4.6. Drugs Interaction

Certain drugs, especially non-steroidal anti-inflammatory agents, such as ibuprofen, may compete with aspirin at the active centre of cyclooxygenase-1 and decrease the effectiveness of the drug [43].

The coadministration of drugs that are also metabolised in the liver by similar cytochrome P450 complex pathways may lead to competition and mutual limitations of the therapeutic effects of the individual drugs. As a result, the antiplatelet effect of clopidogrel may be significantly reduced due to the competitive mechanism of action [44]. Statins and proton pump inhibitors (in particular omeprazole and esomeprazole), the most common types of concomitant drugs used in the secondary prevention of vascular events in addition to antiplatelet agents, may have a potential impact on clopidogrel resistance. Most of these are metabolized via the CYP3A4 and CPC2C19 complex, which can lead to a lower concentration of active clopidogrel metabolites [45]. However, reports regarding this area are conflicting and thus far no clear negative impact has been confirmed [46].

### 4.7. Reduced Bioavailability

There are also factors related to the reduced bioavailability, influencing the response to aspirin. Research demonstrated that non-compliance, non-regular intake, and low doses of aspirin may be associated with a reduced response to the antiplatelet agent [47]. Improving patient compliance or increasing the doses of aspirin (from 75 to 150 mg and up to 300 mg a day) may have a positive effect in reducing the percentage of low responders, leading to “breaking’ the phenomenon of high on- treatment platelet reactivity [48]. Similar dependencies were observed during clopidogrel treatment.

Higher loading doses, such as 600 mg, and higher maintenance dose in critical period of disease may improve platelet inhibitory effect and “break” high on- clopidogrel platelet reactivity [49]. However, an increased bioavailability did not eliminate the variability in clopidogrel response, that confirmed a significant role of different factors contributing to this phenomenon.

The other recognized risk factors for vascular diseases, such as atrial fibrillation, hypertension, ischemic heart disease, chronic heart failure, and alcohol abuse, and the biochemical parameters or markers of inflammatory and prothrombotic processes did not show any significant effect on the low response phenomenon, for both aspirin and clopidogrel. The similarities and differences between factors influencing high on-treatment platelet reactivity, both for aspirin and clopidogrel, are shown in Figure 1.

## 5. Clinical Implications

The mechanisms leading to a low biological response to antiplatelet agents and the incidence of aspirin and clopidogrel resistance are similar, both in ischemic stroke and coronary heart disease. A significant negative impact of the phenomenon on many clinical aspects in both diseases was reported, including a severe clinical course, poor early and late prognosis, an unfavorable clinical outcome, and a higher risk of recurrent vascular events [50,51,52].

Unfortunately, the current standards of management in both units do not take into account the routine assessment of platelet reactivity, which allows for the detection of phenomenon, due to the lack of large clinical trials in this field [1]. This assessment may be considered only in specific situations. For instance in patients who have multiple known risk factors that may be associated with a low biological antiaggregant response. An individualized or personalized approach is necessary. The choice of the method for assessing platelet function is an individual matter of the authors, as no assay has achieved a significant advantage in terms of sensitivity or specificity over others. There is still no accepted standardized methodology in this aspect.

The limited options for antiplatelet therapy distinguish the secondary prevention of ischemic stroke from coronary heart disease. In contrast to coronary heart disease, where prasugrel and ticagrelor are currently recommended, there is still the lack of evidence for prasugrel and ticagrelor in the secondary prevention in ischemic stroke. Thus, if resistance is detected, the patient has more options to switch treatment and replace one drug with another, compared to stroke subjects. In patients with stroke, the switching of antiplatelet agents may not be of benefit to the patient, given the similarities in the mechanisms of resistance to both drugs, resulting in co-resistance in about 50 percent of the cases [4].

Thus, there is still a need for research into new therapeutic options to overcome the low biological response to antiplatelet agents. Ticagrelor was not more effective than aspirin in preventing recurrent ischemic events [53], but the combination of ticagrelor and aspirin was superior to aspirin alone [54] in reducing the risk of vascular events despite the higher risk of severe bleedings. A direct antiplatelet effect was shown for ibrutinib, an irreversible inhibitor of Bruton’s tyrosine kinase that affects the platelet glycoprotein VI signaling and may reduce collagen-mediated platelet aggregation [55]. Glenzocimab, an inhibitor of the platelet glycoprotein VI, involving collagen-induced platelet aggregation, is another option to consider as an antiplatelet agent in the acute phase of ischemic stroke [56]. Studies on the possible usefulness of these drugs in medical practice should be continued, whereas they were not still approved for clinical application.

## 6. Conclusions

In summary, the low biological response to antiplatelet agents is an important and widespread phenomenon distinguished by a multifactorial background. Complex and unexplained mechanisms and etiology underlie aspirin and clopidogrel resistance, which produce a significant reduction on the therapeutic effect of antiplatelet drugs, translating into a worse prognosis of patients with ischemic events. Despite the lack of indications for the routine determination of platelet reactivity to detect a resistance, in the light of the reports on the prevalence and negative impact on the clinical condition, such a determination should be performed in selected patients from the high-risk group. I propose that such an individual approach to specific subjects would enable more effective and efficient secondary prevention, which would be beneficial for patients. 

However, antiplatelet therapy is significantly reduced in maneuverability in ischemic stroke compared to coronary heart disease. Despite completely different metabolic pathways and mechanisms of action for the two most commonly used antiplatelet agents, many similarities and common causes related to this phenomenon have been demonstrated, which may explain the high percentage of coexistence of a low response to both drugs simultaneously. Based on the emphasized differences underlying this phenomenon, such as smoking or concomitant medications, it may be decided in certain clinical situations to include an antiplatelet agent, which will be associated with a lower risk of high in-treatment platelet reactivity. Thus, even without the determination of platelet reactivity, we can increase the chances of effective secondary prevention in ischemic stroke subjects.

## Figures and Tables

**Figure 1 medicina-57-00059-f001:**
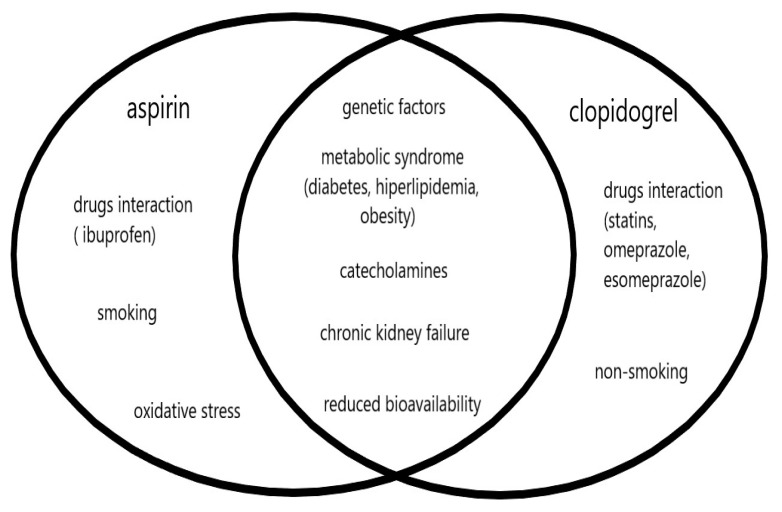
Comparison of factors influencing the decreased responsiveness of platelets during aspirin and clopidogrel treatment.

**Table 1 medicina-57-00059-t001:** Most common platelet function tests and cut-off values discriminating low antiaggregant biological responders.

Platelet Function Assay	Cut-Off Values—Aspirin	Cut Off Values—Clopidogrel
Light transmission aggregometry (LTA)	MAP > 20%	MAP > 46%
Impedance aggregometry (Multiplate)	AUC > 40	AUC > 46
Turbidimetric aggregometry (VerifyNow)	ARU > 550	PRU > 230
Platelet Function Analyzer-100 (PFA-100)	CT < 193 sec.	CT < 106 sec
Vasodilator-stimulated phopshoprotein (VASP)	_	PRI > 50%

AUC—area under the curve units; ARU—Aspirin reaction units; PRU—P2Y12 reaction units; CT—closure time; MAP—maximal platelet aggregation; PRI—platelet reaction index.

## Data Availability

All data presented in this review are included in this article.

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
