# Peer review of "Multifactorial Background for a Low Biological Response to Antiplatelet Agents Used in Stroke Prevention"

_medicina, 2021, doi:10.3390/medicina57010059_

Round 1

Reviewer 1 Report

Title: Multifactorial background for a low biological response to antiplatelet agents used in stroke prevention

Manuscript ID: medicina-1052114

This is an interesting review about to establish the background of resistance to antiplatelet agents used in the secondary prevention of IS and to explain the possible mechanisms.  

The author suggests that genetic, pharmacokinetic, pharmacodynamic factors may influence the incidence of a low biologic response.

In spite of theses attractive reviews, some careful considerations should be made.

Major point
1. More concise but concentrated contents should be needed.

Introduction for multiple factors including genetic, metabolic, and PK/PD is useful and interesting.

But in metabolic disorders related to this phenomenon, some explanations on how diabetes, hyperlipidemia, smoking, obesity and CK , etc. affects are simply described.

I think it is a little superficial.

  1. Why did the authors choose only 2 antiplatelets (aspirin, clopidogrel)?

I agree that these two antiplatelet agents are most commonly used drugs in IS.

But up to date, a few antiplatelet agents are accepted for drugs in secondary stroke prevention.

For example, aspirin, clopidogrel, cilostazol, aspirin+dipyridamole, triflusal, ticlopidine….

When I read the title initially, as a reader, I think I will get the information for every drugs.

  1. The authors presented the platelet function test in Table 1.

In manuscript, please describe about those tests in more detail. (strength, limitation, specificity, sensitiviry?? )

  1. There are too many paragraphs in the manuscript.

Combine paragraphs between related topics. (genetic, PK, PD, metabolic… )

  1. The distinction between PK and PD is ambiguous.

In page 3, line 111-117,

Is it about the pharmacokinetic factors? I think it is related to PD factors more or intermingled. It would be better to distinguish two components clearly.

  1. In pate 5, Line 208,

“According to the updated guidelines, only aspirin and clopidogrel are currently recommended in IS, in contrast to ~~ “

I know that the authors emphasize the lack of evidence for prasugrel and ticagrelor in IS.

But this sentence may confuse the readers that only 2 drugs are recommended in the guidelines.

  1. In page 5, line 215-224,

Ibrutinib and Glenzocimab was not accepted for the secondary stroke prevention yet.

You need to tone down a little.

Minor point

  1. In page 5, line 216,

Ticagrelolà ticagrelor

Reviewer 2 Report

The manuscript is a review of clinical factors related to antiplatelet therapy in the stroke secondary prevention. A low response to antiplatelet agents, such as aspirin or clopidogrel, is a phenomenon that significantly reduces protective effects of the therapy. In the manuscript the Author regards to the factors influencing the incidents of insufficient platelet inhibition during aspirin or clopidogrel therapy in stroke patients. The main determinant described in this review except for genetic disturbances, is high on-treatment platelet reactivity – the phenomenon which the Author studied earlier and reported the results in a few publications (the list of publications is below). 

1: Wiśniewski A, Filipska K. The Phenomenon of Clopidogrel High On-Treatment Platelet Reactivity in Ischemic Stroke Subjects: A Comprehensive Review. Int J Mol Sci. 2020 Sep 3;21(17):6408. doi: 10.3390/ijms21176408.

2: Wiśniewski A, Filipska K, Sikora J, Ślusarz R, Kozera G. The Prognostic Value of High Platelet Reactivity in Ischemic Stroke Depends on the Etiology: A Pilot Study. J Clin Med. 2020 Mar 20;9(3):859. doi: 10.3390/jcm9030859.

3: Wiśniewski A, Filipska K, Sikora J, Kozera G. Aspirin Resistance Affects Medium-Term Recurrent Vascular Events after Cerebrovascular Incidents: A Three- Year Follow-up Study. Brain Sci. 2020 Mar 19;10(3):179. doi:

10.3390/brainsci10030179.

4: Wiśniewski A, Sikora J, Sławińska A, Filipska K, Karczmarska-Wódzka A, Serafin Z, Kozera G. High On-Treatment Platelet Reactivity Affects the Extent of Ischemic Lesions in Stroke Patients Due to Large-Vessel Disease. J Clin Med. 2020 Jan 17;9(1):251. doi: 10.3390/jcm9010251.

5: Wiśniewski A, Sikora J, Filipska K, Kozera G. Assessment of the relationship between platelet reactivity, vascular risk factors and gender in cerebral ischaemia patients. Neurol Neurochir Pol. 2019;53(4):258-264. doi:

10.5603/PJNNS.a2019.0028.

The aim of this review is interesting and important in clinical practice, however the manuscript has been prepared a little bit carelessly. Therefore, the paper in the present form, in my opinion, cannot be recommended for publication – major revision is required.

The Author described in the manuscript in two separate paragraphs (one for aspirin and second for clopidogrel), the similar factors influencing a decreased responsiveness of platelets during treatments with aspirin or clopidogrel (genetics, high platelet reactivity, smoking, diabetes, chronic kidney failure). Maybe, the reorganisation of the manuscript will improve its readability -  the different factors or the groups of factors could be described in the separate paragraphs for both aspirin and clopidogrel. Also, the schema illustrating and/or comparing the different factors could be a good supplement.

The phenomenon of high on-treatment platelet reactivity was mentioned in the manuscript but in my opinion this issue should be described with more attention, especially for aspirin and clopidogrel.

The appropriate references should be included into Table 1 for the cut-off values.

In my opinion, the description of COX-2 inhibition in the platelets should be excluded from the manuscript, because it is nonsignificant issue in antiplatelet therapy.

In my opinion “high antiaggregant platelet reactivity” is not relevant term and not should be used as a replacement for high on-treatment platelet reactivity.

Minor comments

 Line 52, pp.2

Is “The aim of the current study is…”, rather should be The aim of the current review.

Line 118, pp.6

Is …”the definition of high on-aspirin reactivity”, rather should be “the definition of high on-aspirin platelet reactivity”.

Round 2

Reviewer 1 Report

The author has modified the manuscript appropriately for my review.

Reviewer 2 Report

Thank the Author for the responses to my comments from  Report 1.

I accept the new version of the manuscript.